# Congenital Chagas disease: A cohort study to assess molecular diagnostic methods at the Chagas disease national reference center of Argentina

**Constanza Lopez-Albizu**[1] *, **Carolina Inés Cura**[1], **Juan Carlos Ramirez**[2], **Pamela Peyran**[1], **Andrés Benchetrit**[3], **Emmaría Danesi**[1], **Sergio Sosa-Estani**[4,5]

1 Instituto Nacional de Parasitología "Dr. Mario Fatala Chaben", Administración Nacional de Laboratorios e Institutos de Salud "Dr. Carlos G. Malbran", Buenos, Aires Argentina, 2 Instituto Multidisciplinario de Investigaciones en Patologías Pediátricas (IMIPP), CONICET-GCBA, Buenos Aires, Argentina, 3 Hospital F. J. Muñiz, Buenos Aires, Argentina, 4 Centro de Investigaciones Epidemiológica y Salud Pública (CIESP-IECS) CONICET, 5 Drugs for Neglected Diseases initiative, Rio de Janeiro, Brazil

* constanzalopezalbizu@gmail.com

## Abstract

### Background

*Trypanosoma cruzi* is a protozoan parasite which causes Chagas disease. Mother-to-child transmission is the main route of transmission in vector-free areas. Congenital Chagas disease refers specifically to cases arising from this route of transmission. This work evaluates the clinical sensitivity of two qPCR techniques for diagnosis of congenital Chagas disease.

### Methods

The study was developed in the National Institute of parasitology (NIP), Argentina, and Pan-American Health Organization/ Word Health Organization Collaborating Center for Chagas Disease. Between July 2014 and May 2018, a prospective cohort study was carried out with 499 children born to seropositive for *T. cruzi* infection included. The performance of qPCR techniques was compared with the gold standard diagnostic algorithm for Congenital Chagas disease (CCD-GS), which comprises performing more than one parasitological test on children from birth until nine months of age, and serology from ten months of age.

### Findings

Of the 961 babies born to women seropositive for Chagas disease who were attended at the NIP laboratory, 462 did not meet the study inclusion criteria; 22 cases were diagnosed with congenital Chagas disease. qPCR showed 100% clinical sensitivity and 98 to 100% clinical specificity for the diagnosis of congenital Chagas disease compared with CCD-GS algorithm.

**Data Availability Statement:** All relevant data are within the manuscript and its Supporting Information files.

**Funding:** The author(s) received no specific funding for this work.

**Competing interests:** The authors have declared that no competing interests exist.

## Interpretation

The results obtained in this study demonstrate the clinical accuracy and effectiveness of qPCR SatDNA and qPCR kDNA for diagnosis of congenital Chagas disease. It could be a powerful tool for chagas test and treat strategies to reduce late complications of the disease.

## Funding

This work was financed by the INP Dr. Mario Fatala Chaben, ANLIS Dr. Carlos G. Malbran.

### Author summary

*Trypanosoma cruzi* is a protozoan parasite which causes Chagas disease. Mother-to-child transmission is the main route of transmission in vector-free areas. Congenital Chagas disease refers specifically to cases arising from this route of transmission. Diagnostic for Congenital Chagas disease comprises performing more than one test on children from birth until nine months of age, to ten months of age. This work evaluates two qPCR techniques for diagnosis of congenital Chagas disease with high sensitivity and bring treatment during the period in which tripanocydal drugs has high efficacy.

## Introduction

*Trypanosoma cruzi* is a protozoan parasite which causes Chagas disease. Based on genetic differences within the *T. cruzi* species, seven discrete typing units (DTUs) (TcI to TcVI, and TcBat) have been defined [1–3]. *T. cruzi* infections have an acute phase lasting three to four months with high parasitemia, followed by a chronic phase with intermittent low parasitemia. In the chronic phase, specific antibodies are detected and 30% of people with *T. cruzi* develop clinical manifestations, the most frequent being cardiac and gastro-intestinal [4–7]. Mother-to-child transmission, from which a transmission rate of 5% is reported [8] is the main route of transmission in vector-free areas. Congenital Chagas disease (CCD) refers specifically to cases arising from this route of transmission. Given the low frequency of clinical manifestations in the acute phase, diagnosis is made using laboratory tests (CCD-GS algorithm). Due to the low diagnostic sensitivity of parasitological methods, such as IgM detection, and the persistence of maternal antibodies in infants up to 9 months of age, the CCD-GS algorithm involves performing more than one parasitological test on children from birth until nine months of age, and serology from nine to ten months of age (S1 Fig) [4,5]. Although this diagnostic algorithm has high diagnostic sensitivity and specificity, unfortunately the whole protocol (including serology) is completed in only 12% of cases in healthcare centers and 55% in reference centers [9–11]. Treatment with trypanocidal drugs is safe and has high efficacy in neonates and during the first life year [4,12]. Since therapeutic success is inversely proportional to the age at which treatment begins, early diagnosis and treatment would improve the prognosis and chance of cure [13].

 PCR tests have been used to detect *T. cruzi* DNA in peripheral blood samples since the early 1990's and since then have been used in research studies and clinical trials on CCD [14]. Real-time quantitative PCR (qPCR) has also been used with promising results for diagnosis [15,16].

There is, however, variability in the test characteristics of qPCR for CCD diagnosis depending on the epidemiological characteristics of the population studied, the volume of sample collected, and the methods and equipment used for analysis. In 2007, recognizing the need for a standardized protocol, a multicenter study was developed where the same samples were analyzed with different routinely used PCR techniques. Subsequently, a workshop was held to standardize PCR protocols [17]. The next step was the analytical validation of two qPCR tests for the quantification of *T. cruzi* [18,19]. In a study of PCR techniques for CCD diagnosis carried out by teams from the National Institute of Parasitology, Dr. Mario Fatala Chaben (NIP), the Argentinian National Chagas Reference Center, and a PAHO-WHO collaborating center, PCR techniques showed 100% diagnostic sensitivity and specificity [20].

Given the need to implement a laboratory method with greater diagnostic sensitivity, while maintaining high specificity, in 2014, the NIP began a trial for implementing qPCR for the diagnosis of CCD within the national public health system. The first phase of implementation was an evaluation of PCR techniques. With the aim of reducing the risk of carryover contamination uracil-DNA glycosylase (UNG, Thermo Fisher Scientific, Rockford,IL) was added to the reaction mix. And, to increase availability and access for smaller laboratories, using a commercially available internal amplification control was used instead of a "*In-house*" option [21]. Analytical verification, following international guidelines, was carried out [21]. The calculated limit of detection [the lowest parasitic load that gives 95% of positive results (LOD95%) was 0.87 par. eq./mL (95% CI, 0.62–1.24 par.eq./mL) for the SatDNA qPCR and 0.43 par. eq./mL (95% CI, 0.32–0.59 par. eq./mL) for the kDNA qPCR [21].

The current study was designed to evaluate the clinical sensitivity of three diagnostics algorithm for CCD using combinations of two *in-house* qPCR techniques for SatDNA and kDNA targets, compared with the current standard [21].

## Materials and methods

### Ethical statement

The study was approved by the National Institute of parasitology Dr. Mario Fatala Chaben Bioethics Committee with the approval number 2–2018. The samples were collected and additional data was requested during patients visits to the NIP to diagnose congenital Chagas disease to ensure informed participant involvement. Verbal consent was obtained from all participants and/or parents or guardians of child participants. Additionally, all data and samples were coded to anonymize patient identities and protect confidentiality.

The ethics committee recomended implementing written consent procedures retrospectively (S1 Written Consent Form) This was feasible in most patients (413 patients–83%). We assumed the oral consent was enough based on the fact that this practices were routinely performed. Additionally, in cases of confirmed CCD, medical recommendations and trypanocidal treatment were provided.

### Study population

A prospective cohort study was carried out. Children born to mothers with *T. cruzi* infection, confirmed in accordance with the standard diagnostic of PAHO guidelines, were invited to participate. These guidelines consider that a person is infected when their blood has reactive results with at least two of the following tests using different principles and antigens: enzyme-linked immunosorbent assay (ELISA)-indirect agglutination assay (IHA), ELISA-immunofluorescence (IIF), or IHA-IIF. The two tests must be carried out in parallel and, in the event that only one is reactive, a third test that has not already been used should be performed [4,10,11,22]. People seeking a diagnosis from the diagnostics department of the NIP, Buenos

Aires City, between July 2014 and May 2018 were invited to participate. In order to demostrate that the diagnosis of CCD with qPCR techniques were no inferior to CCD-GS algorithm, using the Buderer method [23] for an alpha level of 0.05, estimating a congenital transmission rate of 3.5%, expecting a sensitivity and a specificity of 95%, with an acceptable width of the 95% confident interval (CI) of no more than 10%, and awaiting a dropout rate of 15%, 829 participants would need to be screened, and 615 recruited. Babies of less than ten months old whose mothers had *T. cruzi* infection confirmed by standard diagnostics and who had completed the full CCD-GS algorithm were included for the final analysis. Babies whose mothers had standard diagnostic tests negative for *T. cruzi* infection, infants over ten months old, and infants with difficult venous puncture were excluded. In cases where the blood sample was difficult to obtain and therefore, a low volume was obtained, the child was excluded from the study for ethical reasons, in order to avoid performing a new puncture. The small volume obtained was used for the reference diagnosis (CCD-GS) algorithm.

## Sample collection

Samples were taken from infants at 15 days (T1), three months (T2), and ten months after birth (T3). Venous blood (VB) samples were taken and aliquoted immediately for conservation using different methods. **Real time polymerase chain reaction (qPCR) performed at T1 and T2:** 0.5 mL samples stored with an equal volume of 6 M guanidine hydrochloride Buffer 0.2 M EDTA (Sigma-Aldrich, St. Louis, Missouri, United States) (GEB) for 72 hours at 25°C then 4°C. **Micromethod (MM) performed at T1 and T2:** 0.5 mL samples stored with 20 μL of sodium heparin at 25°C (2 to 4 hours). **Standard diagnostic (SD) performed at T3:** 1 mL samples stored with separator gel and coagulation accelerator at 4°C (24 to 48 hours).

## Study design

To evaluate qPCR in the diagnosis of CCD-GS, qPCR was performed on T1 and T2 samples in parallel to the MM. qPCR tests were performed in two steps.

**Step 1:** DNA extraction, single reaction SatDNA and kDNA qPCR (Figs 1 and 2).

**Step 2:** In detectable samples, a second DNA extraction and qPCR SatDNA and kDNA were performed in duplicate (Fig 2).

When the baby's first qPCR was detectable, the family was contacted and asked to come to the INP for further analysis on a new sample. The subsequent samples were used to perform MM, qPCR, and, if the baby was ten months or older, a standard diagnostic, thus confirming CCD according to the CCD-GS algorithm.

Families of children with CCD were contacted and the children were given medical care and the necessary treatment.

## Data analysis

Categorical variables were expressed as category frequencies and percentages, while continuous variables were presented as mean and standard deviation or median and range. CCD prevalence in the studied population, as well as sensitivity, specificity, and positive and negative predictive values for each algorithm, were expressed in percentages with confidence intervals of 95%. Continuous variables were compared with Welch´s unequal variance t-test. Receiver operating characteristic (ROC) curves and the corresponding areas under the curves (AUC) were carried out to assess the accuracy of the methods. Multiple groups means where compared with one way ANOVA, Tukey´s HSD was used for post hoc analyses. The assumptions underlying the statistical analyses were evaluated and met. Statistical analyses were done with R version 4.3.0.

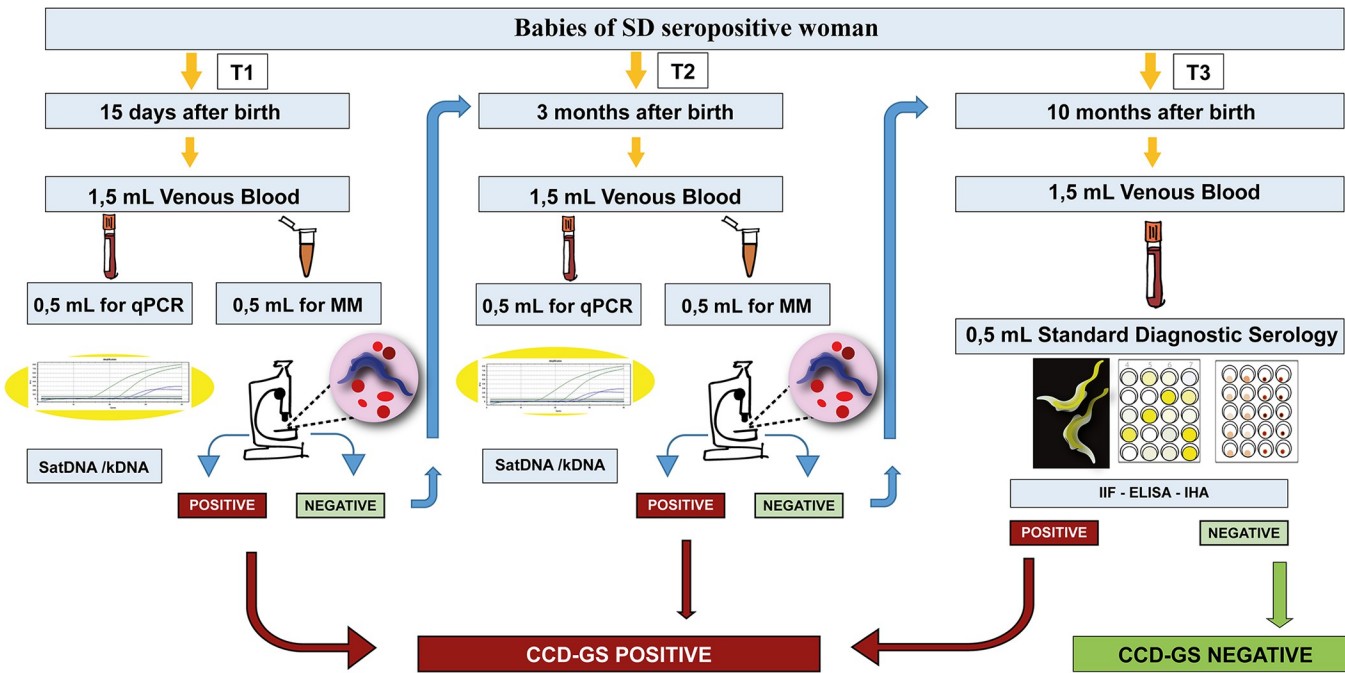

**Fig 1. Study flow chart; SatDNA: Satellite DNA qPCR; kDNA: kinetoplast DNA qPCR; CCD: Congenital Chagas Disease; MM: micromethod.**

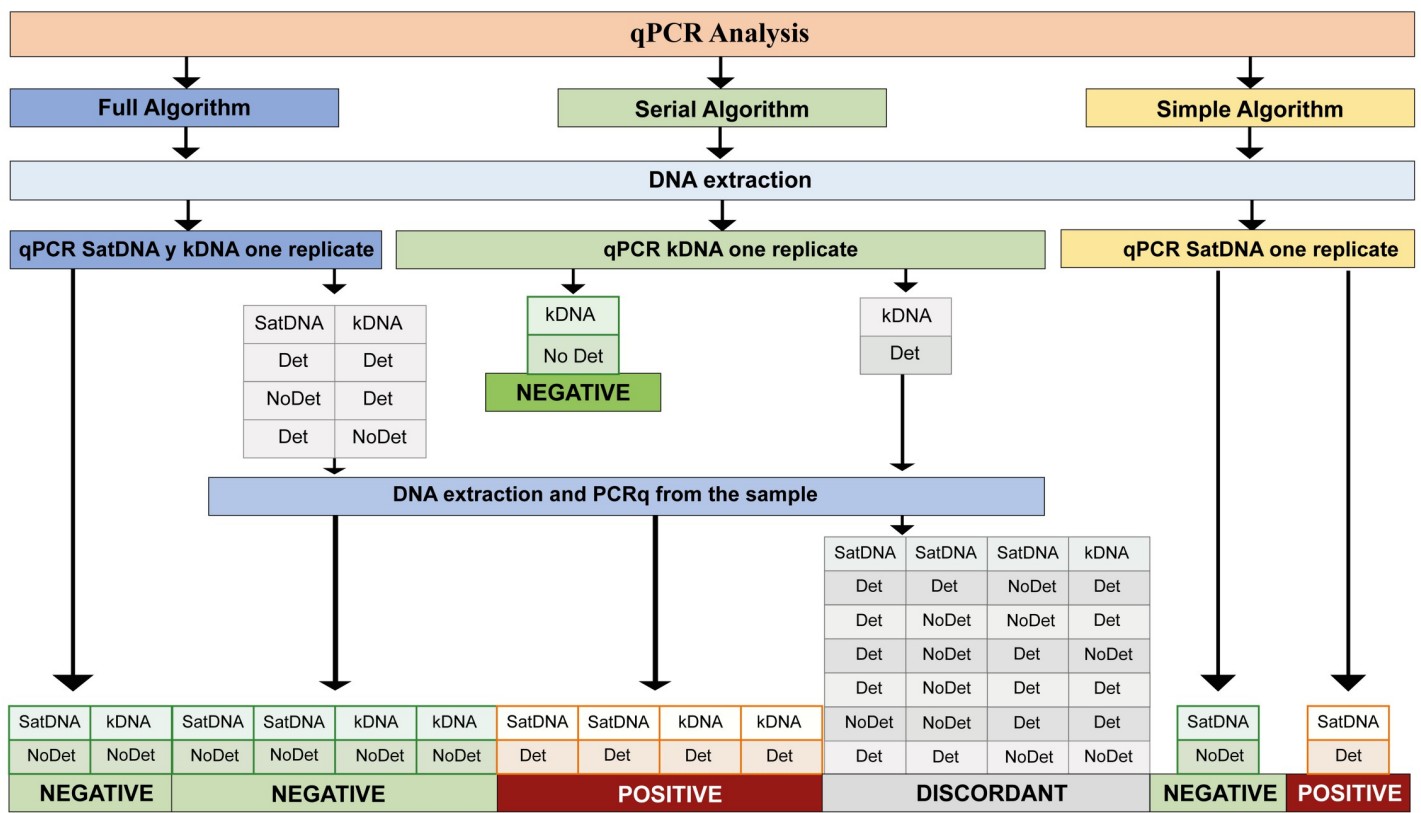

**Fig 2. Flow chart of the trial inclusion algorithm.** SD: Chronic Chagas Disease Standard Diagnostic; CCD-GS: CCD gold standard diagnostic algorithm. T1, T2, and T3: samples taken at 15 days (T1), 3 months (T2), and 10 months after birth (T3).

The performance of SatDNA and kDNA qPCR, as measured by sensitivity, specificity, positive predictive value, and negative predictive value, was compared to the CCD-GS algorithm.

Three diagnostic algorithms for qPCR were analyzed: Full algorithm, Serial Algorithm, and Simple Algorithm (Fig 2). All data sets are available at: https://data.mendeley.com/preview/x36vmxdv6h?a=9896c8e6-d347-4659-b37e-8e50ca6ebb92.

## Reference standard according to the current CCD diagnostic algorithm

The CCD-GS algorithm diagnosis was performed following national guidelines [4,10]. Current diagnosis of CCD is based on positive parasitological tests shortly after birth and/or serological reactivity after nine months of age. Parasitological techniques detect motile parasites in peripheral blood, concentrate parasites by centrifugation using microcentrifuge tubes (Micromethod (MM)) followed by microscopic examination of the buffy coat. If the parasitological test is negative at birth, it can be repeated at one to three months of age, when the peak of parasitemia is usually observed (S1 Fig). **Parasitological method:** Parasitological diagnosis was made with the Micro-method (MM) described by *De Rissio et al.* 2010 0.5 mL of anticoagulated venous blood (heparin) was centrifuged for one minute at 3000 r.p.m. Two drops from the buffy coat, between the plasma and the blood cells, were extracted with a pipette. In a slide topped with two 22mm x 22mm coverslips, each droplet was observed completely in greek guard form [10]. **Standard diagnostic:** was performed at the NIP with "*in-house*" serologic methods. The *in-house* tests were developed with the following antigens: i) enzyme-linked immunosorbent assay (ELISA): soluble fraction of epimastigote lysate of *T. cruzi* strain Tul2, ii) indirect hemagglutination (IHA): epimastigote lysate of 29 *T. cruzi* strains, iii) indirect immunofluorescence (IIF): whole epimastigotes of *T. cruzi* strain Tul2 preserved in formaldehyde. The cut-off values for the techniques are as follow: i) ELISA: OD$\geq$(average of high positive controls + average of low positive controls) x 0.28; ii) IHA: reactive title$\geq$1/32; iii) IIF: reactive title$\geq$1/32. Internal quality controls were used in all serological assays. These controls were prepared using pools of recalcified plasma from positive and negative patients obtained from blood banks in Argentina. The tests were performed following the standardized procedures of the NIP and under an external quality control program by the Brazilian Society of Clinical Assays [24]. Each serological tests were performed once per sample. If only one assay results reactive all there assays were repeated once on the same serum sample. The mothers were classified as positive based on their serological status, which showed reactivity in two out of the three assays (ELISA, IHA, or IIF).

## Real time PCR assays

**DNA extraction:** DNA was extracted from 300 µL of samples stored in GEB using the High Pure PCR Template Preparation kit (Roche Diagnostics GmbH, Mannheim, Germany) and eluted in 100 µL of elution buffer, as described by *Duffy et al.* 2009 [25]. Extracted DNA was stored at -20˚C until used for PCR [21]. **qPCR procedures:** In accordance with *Cura et al.* 2017 [21] two previously analytically validated qPCR procedures were performed in a NIP diagnostic laboratory: satellite DNA (SatDNA) and kinetoplast DNA (kDNA). **SatDNA:** amplifies the satellite sequence from *T. cruzi* nuclear DNA. The satellite DNA based primers and probe sequences are primer cruzi1 forward: 5'-ASTCGGCTGATCGTTTTCGA-3, primer cruzi2 reverse 5'-AATTCCTCCAAGCAGCGGATA-3', and cruzi3 probe FAM-5'-CACACAC TGGACACCAA-3'-NFQ-MGB. And the human RNase P gene, as an endogenous amplification control, using the TaqMan RNase P Control. **kDNA:** amplifies the amplified kDNA. The kDNA based primers and probe sequences are primer 32F forward 5'-TTTGGGAGGGGCGT TCA-3', and primer 148R reverse 5'-ATATTACACCAACCCCAATCGAA-3', and 71 probe

FAM-5'-CAT+CTCA+CC+CGTA+CATT-3'-BHQ1 (the + in front of the nucleotide indicates a locked nucleic acid monomer substitution. And the human RNase P gene, as an endogenous amplification control. Both qPCR procedures using the TaqMan RNase P Control Reagents Kit (Applied Biosystems, Foster City, CA) at a final concentration of 0.5X and were performed with 5 µL of extracted DNA, using FastStart Universal Probe Master Mix (Roche Diagnostics GmbH, Mannheim, Germany) in a final volume of 20 µL, using an ABI7500 (Applied Biosystems) and a CFX96 (BioRad) qPCR device. Cycling conditions for both qPCR assays were a first step of 2-minute hold step at 50˚C, 10 minutes at 95˚C, followed by 40 cycles at 95˚C for 15 seconds and 58˚C for 1 minute. A sample was considered positive for *T. cruzi* DNA when the amplification curve crossed the fluorescence threshold generating a quantification cycle (Cq) <40 value using a cycle threshold of 0.01 for kDNA; 0.02 for SatDNA in Abby 7500 thermocycler and 40 for both qPCR in a CFX96 Thermocycler [26]. **Quality Control Assessment of qPCR:** As a contamination control in DNA extraction, seronegative human blood samples preserved with GEB were used for every 11 patient samples. Seronegative human blood samples contaminated with CL Brenner strain epimastigotes at a concentration of 1 parasite per ml of blood was used as a positive control. And a no template control was used in qPCR reaction.

## Quantification of blood parasitic loads

After the DTUs of the positive cases were known, the quantification of the samples was carried out. Given the variability in satellite DNA repeat sequences between different DTUs, and the fact that parasite DTU identification showed TcV results in children with CCD, seronegative human blood samples were spiked with $10^5$ par. eq. /mL of LL014-1-R1 Cl1 *T. cruzi* stock (TcV) cultured epimastigotes and mixed with an equal volume of GEB. The number of parasites was determined using a hemocytometer and verified in a Z2 Coulter particle count and size analyzer. DNA extraction was performed as described by *Duffy et al.* 2009 [25]. Quantification of positive samples was performed using a standard calibration curve (measured values). The standard curve was constructed by serially diluting total DNA extracted from non-infected human samples across a concentration range of $10^5$ to 1 par.eq./mL for quantification of blood parasitic loads via SatDNA qPCR. The curve was validated based on an efficiency of 90–100% and an r-squared value of 0.9–1.

## Parasite DTU identification

*T. cruzi* DTU identification was done using a sequential algorithm of three multiplex real-time PCR Assays (SL-IR_MTq, 18S-COII_MTq, and 24Sα_MTq), as previously described [27].

# Results

## Participants included in the study

The participants enrolled in this study are shown in Fig 3. Of the 961 screened, 829 were enrolled, and 499 completed the follow-up.

The transmission rate of CCD in the infant population observed at the NIP from 2014 to 2018 was 4.8% (95% CI = 3.13–6.47; 30/631), while the rate of CCD in babies included in this study was 4.4% (95% CI = 2.6–6.2; 22/499).

The median age of the whole population (N= 499) at time of the first sample was 1.8 months (range 0–9.4 months), while for the subpopulation with CCD, this 2.1 months (range 0.5–9.0 months). Biological sex distribution was: CCD-GS algorithm positive cases: 45% male, 55% female, CCD-GS algorithm negative cases: 50% male, 50% female (Table 1).

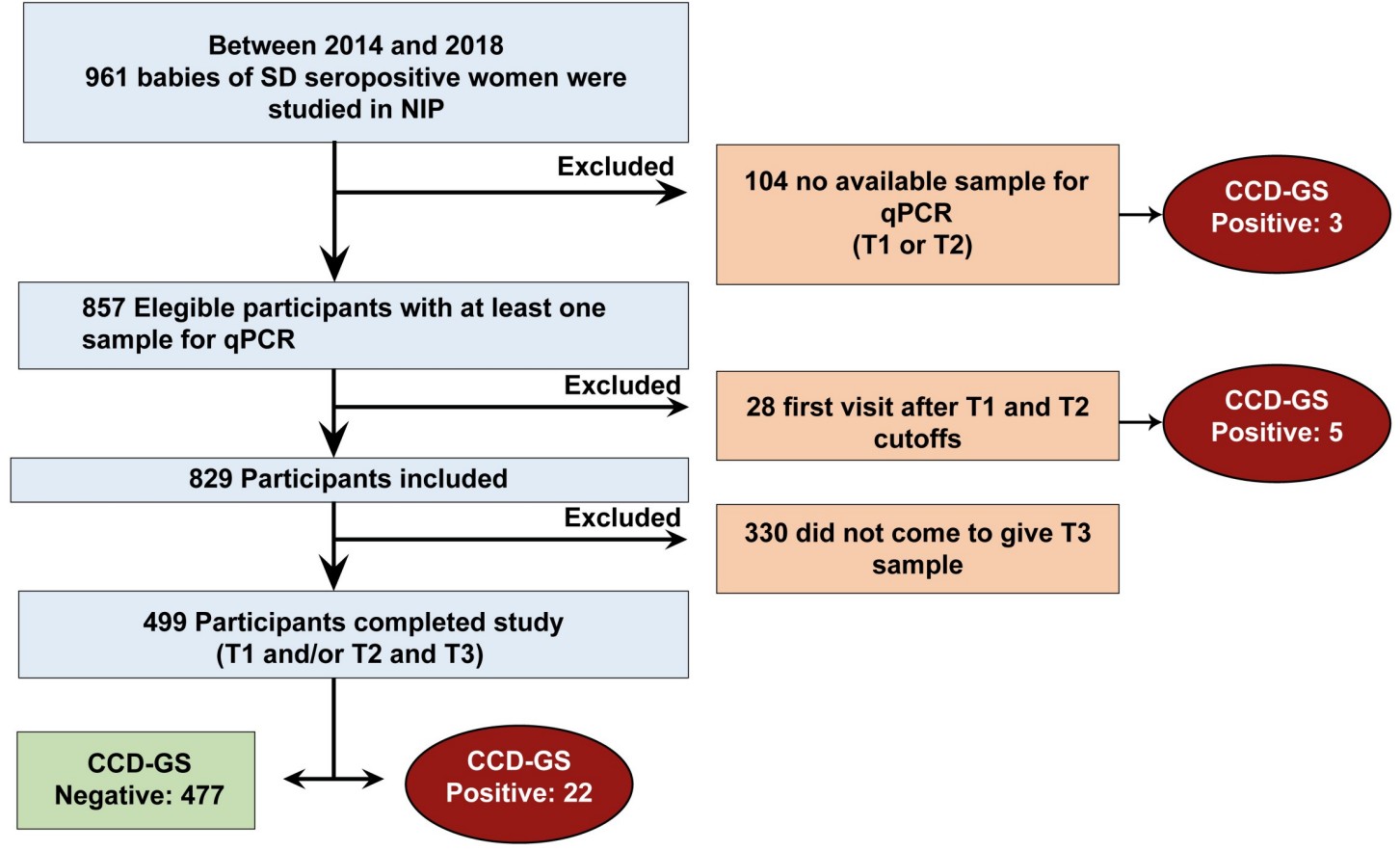

**Fig 3. Diagnostic algorithms analyzed for the diagnosis of CCD with qPCR test.**

### Index test

**Step 1.**   In Step 1, both qPCR methods (SatDNA and kDNA) detected 22 of the 22 cases diagnosed by CCD-GS algorithm in the first sample, while MM detected 11 (50%) of these cases. In addition, qPCR SatDNA gave 4 false positive results, while qPCR kDNA gave 27 false positive results. Of the false positives detected by SatDNA and kDNA qPCR, only 3 cases were positive for both qPCR tests in the same sample. Both qPCR methods (SatDNA and kDNA) showed 100% sensitivity. The specificity observed was 99% (95% CI: 98–100) for qPCR SatDNA, and 94% (95% CI: 92–97) for kDNA.

The $C_q$ of false positive cases (kDNA mean= 36.95, SD= 2.12; SatDNA mean= 37.18, SD= 2.11) was higher than $C_q$ of true positive cases (kDNA mean= 19.92, SD= 2.86; SatDNA mean= 20.03, SD= 2.45), in both qPCR tests this difference was statistically significant (for kDNA t(37.92)= 23.24, p-value < 0.001; for SatDNA t(5.34)= 16.45, p-value <0.001).

The AUC of the ROC curve for both kDNA and SatDNA was 1 (95%CI 1–1) when a Cq threshold of 29.5 for sDNA, and 28.2 for kDNA, was selected; the sensitivity and specificity was 100% for both qPCR tests (S2 Fig).

**Step 2** Table 2 shows the estimated statistical parameters for three diagnostic algorithms (Fig 2) for congenital Chagas disease using qPCR.

DTU identification found TcV in 22/22 CCD-GS positive cases. Parasite quantification and the age of positive test using the CCD-GS diagnostic are shown in Table 3. A one-way

**Table 1. Descriptive characteristics of the population of babies born to SD seropositive women studied at NIP between 2014 and 2018.**

| Variable | N | CCD-GS Positive | CCD-GS Negative |
|---|---|---|---|
| **Babies of SD seropositive mothers studied at NIP and completed CCD-GS algorithm** | **631** | **30 (4.8%)** | **601(95.2%)** |
| **Mothers treated before pregnancy** | | | |
| Yes | **21** | **0** | **21(100%)** |
| No | **610** | **30 (4.9%)** | **580 (95.1%)** |
| **Sex:** | | | |
| Female | **322** | **18 (5.6%)** | **304 (94.4%)** |
| Male | **309** | **12 (3.9%)** | **297 (96.1%)** |
| **Birth weight:** | | | |
| Weight less 2500g | **150** | **14 (9.3%)** | **136 (90.7%)** |
| Weight more tan 2500g | **478** | **16 (3.3%)** | **462 (96.7%)** |
| Not available | **3** | **0** | **3** |
| **Childbirth:** | | | |
| Vaginal | **348** | **19 (5.5%)** | **329 (94.5%)** |
| Caesarean section | **278** | **11 (3.9%)** | **267 (96.1%)** |
| Not available | **5** | **0** | **5** |
| **Gestation:** | | | |
| Preterm-birth | **37** | **4 (10.8%)** | **33 (89.2%)** |
| Full-term-birth | **584** | **26 (4.5%)** | **558 (95.5%)** |
| Not available | **10** | **0** | **10** |

N: numero de casos, CCD-GS: Congenital Chagas disease gold standard algorihm

ANOVA revealed a significant effect of age group (less than 1.5 months, between 1.5 and 4 months, and older than 4 months) on parasitic load (expressed as logarithm): $F_{(2, 19)} = 3.724$, p-value= 0.0432. The effect size ($\eta2$) was 0.282. Tukey's HSD post hoc test showed that children between 1.5 and 4 months of age had significantly higher parasitic loads (expressed as

**Table 2. Results obtained from the application of parasitological methods and the three qPCR diagnostic algorithms on the samples taken at T1, in a cohort of 499 babies of SD seropositive women studied at the INP between 2014 and 2018, compared with the CCD-GS algorithm.**

| Test | Prevalence according to CDD-GS % (95%CI) | Index Text Results | Sensitivity% (CI 95%) | Specificity% (CI 95%) | PPV (CI 95%) | NPV (CI 95%) | Kappa index |
|---|---|---|---|---|---|---|---|
| **Micromethod** | | TP: 11 /22<br>FN: 11/22<br>TN: 11/477 | 50 (28–72) | 100 (99–100) | 100 (72–100) | 98 (96–99) | * |
| **Simple Algorithm** | 4,41<br>(2.6–6,2) | TP: 22/22<br>FP: 4/477<br>TN: 473/477 | 100 (85–100) | 99 (98–100) | 85 (65–96) | 100 (99–100) | 0.933 |
| **Serial Algorithm** | | TP: 22/22<br>FP: 0/22<br>TN:477/477 | 100 (85–100) | 100 (99–100) | 100 (85–100) | 100 (99–100) | 1 |
| **Full Algorithm** | | TP: 22 /22<br>FP: 0/22<br>TN: 477/477 | 100 (85–100) | 100 (99–100) | 100 (85–100) | 100 (99–100) | 1 |

TP: true positive, TN: true negative, FP: false positive, FN: false negative, PPV: Positive predictive value, NPV: negative predictive value, CI: confidence Interval, SD: standard diagnostic, CCD-GS: diagnosis made using laboratory tests. * Since the MM is included at of the GS-CCD algorithm, the calculation of the kappa index calculation between these two methods is not applicable.

**Table 3. Parasite quantification on blood, age at time of the first sample (T1) on babies with CCD included in the study, and age when the test using the CCD-GS algorithm was positive.**

| Positive case N° | Sample | Age (months) | Parasitic Load (par Eq. /mL) | Sample | Age (months) | CCD-GS test |
|---|---|---|---|---|---|---|
| 1 | T1 | 2.0 | 170 576 | T3 | 10.2 | SD |
| 2 | T1 | 2.1 | 2 237 | T2 | 4.2 | MM |
| 3 | T1 | 1.1 | 568 | T2 | 3.3 | MM |
| 4 | T1 | 3.0 | 29 330 | T1 | 3.0 | MM |
| 5 | T1 | 1.2 | 2 507 | T2 | 7.2 | MM |
| 6 | T1 | 1.7 | 8 591 | T1 | 1.7 | MM |
| 7 | T1 | 2.2 | 29 692 | T1 | 2.2 | MM |
| 8 | T1 | 9.0 | 7 149 | T2 | 12.4 | SD |
| 9 | T1 | 3.7 | 72 167 | T2 | 10.3 | SD |
| 10 | T1 | 0.7 | 8 163 | T4 | 9.8 | SD |
| 11 | T1 | 4.0 | 27 171 | T1 | 4.0 | MM |
| 12 | T1 | 1.5 | 23 560 | T2 | 2.6 | MM |
| 13 | T1 | 3.2 | 37 438 | T1 | 3.2 | MM |
| 14 | T1 | 1.1 | 8 349 | T2 | 2.1 | MM |
| 15 | T1 | 4.9 | 72 549 | T1 | 4.9 | MM |
| 16 | T1 | 1.8 | 37 232 | T1 | 1.8 | MM |
| 17 | T1 | 1.1 | 1 113 | T1 | 1.1 | MM |
| 18 | T1 | 2.1 | 46 793 | T1 | 2.1 | MM |
| 19 | T1 | 6.4 | 390 | T4 | 12.0 | SD |
| 20 | T1 | 2.3 | 6234 | T1 | 2.3 | MM |
| 21 | T1 | 0.5 | 29 187 | T2 | 3.2 | MM |
| 22 | T1 | 6.3 | 11 276 | T1 | 6.3 | MM |

MM: Micro Method, SD: Standard Diagnostic, CCD-GS: Congenital Chagas Disease Gold Standard algorithm, CCD-GS test: CCD was confirmed by this test of CCD-GS algorithm.

logarithm) than children younger than 1.5 months, p-value= 0.0434, $d$= 0.8111. When compared with children older than 4 months the group aged between 1.5 and 4 months had a higher parasitic load (expressed as logarithm) but this difference was not significant, p-value= 0.293, $d$= 0.554.

## Discussion

The current gold standard used for CCD diagnosis (CCD-GS algorithm) has low sensitivity in the first 9 months of life, when only parasitological tests are able to confirm diagnosis. To obtain reliable results, more than one test is generally required. In some cases, a first negative parasitological (MM) test results in a definitive negative diagnosis of CCD, which ignores the low sensitivity of this screening method and the need for a serological reference standard once the baby reaches 10 months of age [10,13]. Consequently, in attempts to increase diagnostic sensitivity, in many maternity units three venous punctures are performed on the baby within a week, which is a big stress for both the baby and the family, and that affects negatively for continuity of the CCD-GS algorithm.

Molecular methods are promising alternative methodologies for early detection; hence their use is growing, but they do have some limitations [13].

This study evaluated the performance of two qPCR tests for CCD diagnosis in a cohort of 499 children born to mothers with *T. cruzi* infection. We observed a CCD rate within the

range described in studies from different geographic zones, which range between 2.9% and 11.3% [10,28–37]. This is in agreement with a meta-analysis which found a rate of 4.6% in studies which used the CCD-GS algorithm [8].

Sixty-five percent of the cohort completed the CCD-GS algorithm, in concordance with previous studies at this institute and in Bolivia [20,38]. Completion of the CCD-GS algorithm was notably higher, compared to the 6% to 10% cases tracked annually by the National Program of Epidemiological Tracking (Sistema Nacional de Vigilancia) [39]. These differences can be explained by the pro-active approach taken (calls, explanation talks, orientation) by the INP to reach the children's parents after the T1 sample was taken.

In Step 1 of this study, we observed a higher sensitivity of both qPCR methods for CCD diagnosis compared to the CCD-GS algorithm, advancing the diagnosis by an average of 4.3 months. However, we observed a lower number of false positive results using qPCR SatDNA compared to qPCR kDNA. Similarly, the average $C_q$ of the false positive cases by qPCR was significantly higher than the true positive qPCR results in CCD-GS algortihm positive children, for both kDNA and SatDNA. These false positive results could be differentiated from true positives when a cut-off value was applied, as seen in Benatar *et al.* 2021 (S2 Fig) [28].

All qPCR algorithms had a higher sensitivity than MM (Table 2). This reflects a higher probability that the resulting test will be positive in the case of a child who has CCD, and thus the case can be detected in the healthcare system at an earlier stage of infection [10,20,37,38,40–42], in fact, all the positive cases had detectable qPCR results in the T1 sample (Tables 3 and S1), giving an opportunity for diagnosis and early treatment.

MM yielded 100% specific results while using a direct observation method. The qPCR SatDNA simple algorithm also had a value near to 100%, which indicates this method is sufficient for consideration as a CCD diagnostic method. Consequently, when a child has non-detectable qPCR results in their T1 sample, there is a very low chance of CCD.This has also been observed in previous studies, with a high qPCR negative predictive value (NPV) in children without CCD [20,38,40,43]. Furthermore, the Kappa index for the overall agreement level was almost perfect. In addition, analysis of the DTU of the cases diagnosed with CCD resulted in 100% TcV, which is consistent with previous results in the region [27,28,32].

Analysis of blood parasite quantification in CCD-GS positive samples, showed that a greater burden was observed in children between 1 and 3 months of age. This data is consistent with studies in similar patients [28,29,44].

Given the absence of vectors in the area of the study and 100% control of transfusion centres, propagation of infection is mainly due to vertical transmission from one generation to the next [8]. Given a low rate of completion of the CCD-GS algorithm, the resulting lack of diagnosis in children, and the reduced effectiveness of treatment after the first year of life, it is essential to enhance the performance of the methods used to detect new cases of CCD as an intervention with a high impact on public health [45–50]. This study was conducted in a reference laboratory, employing *in-house* methodologies and utilizing a non-commercially available preservative (GEB) not pre-loaded in ready-to-use tubes.

As a possible application goal of this study is to advance the availability of usable diagnosis methodologies for detecting CCD with minimum impact on the affected families, we recognize the methodologies and necessary equipment must be brought to an accessible level for national application, as many laboratories across the country have limitations in their access to used materials. Further studies need to be done to test the possible application of standardized diagnostic kits which have recently become commercially available. Additionally, the *in-house* methods need to be tested within a context of the flexibility of application to ensure each laboratory can achieve the same accuracy within the approach detailed in this study with the

constraints of the materials and thus methodologies they have access to implement in their regions.

This study sets a precedent for clinical implementation of qPCR for CCD diagnosis. These results, along with two similar studies, establish a basis for implementation of qPCR in the Argentinian CCD diagnosis algorithm [28,29,51].

## Conclusion

The results obtained in this study demonstrate the clinical accuracy and effectiveness of qPCR SatDNA and qPCR kDNA tests for CCD diagnosis. Likewise, in 50% of the children with CCD having a negative MM, but *T. cruzi* DNA detectable by qPCR, the infection was confirmed with CCD-GS, requiring one to two additional visits and samples (T2 and/or T3). Although MM is the only test widely available, due to its simplicity, it has been shown to have low sensitivity for CCD diagnosis compared to qPCR. This unique approach may pose challenges for implementation in healthcare system laboratories. Nevertheless, qPCR has been shown to have great potential for the diagnosis of CCD diagnostic during the first three months of life, avoiding loss before completion of serology and allowing access to early and effective treatment.

## Supporting information

**S1 Fig. CCD Gold Standard Algorithm for CCD Diagnosis Foot figure: CCD: congenital Chagas disease, MM: micro method.**
(TIF)

**S2 Fig. ROC curves and box plots obtained from the $C_q$ values analysis from SatDNA and kDNA qPCR results on sample T1, in children under 10 months old in a cohort of 499 children of seropositive mothers between 2014 and 2018 during the implementation trial at the NIP Dr.** Mario Fatala Chaben
(TIF)

**S1 Written Consent Form. Written Consent Form for written consent procedures implemented for this study.**
(PDF)

**S1 Table. Quantitative PCR results (Cq values), parasitological and serological methods, and parasite load quantification in samples collected at time points T1, T2, and T3 from children with confirmed congenital Chagas disease".** Foot table: W: well, S: SatDNAqPCR, k: kDNA qPCR, MM: Parasitological test by Micromethod, SD: standard diagnostic results, R: reactive, NR: no reactive.
(DOCX)

## Acknowledgments

We thank the Diagnostic department team for their collaboration: Daniela Oliveto, Laura Mansilla, Mariana Gosende, Marisa Pico, Oscar Fernandez, Marcelo Gonzalez, Indira D'Amico, Victoria Andrade and Claudia Nose for the editing of the figures, and Maria Soledad Santini for her support. DNDi is grateful to its donors, public and private, who have provided funding to DNDi since its inception in 2003. A full list of DNDi donors can be found at www.dndi.org/about/donors/our-grants.

## Author Contributions

**Conceptualization:** Constanza Lopez-Albizu, Carolina Inés Cura, Juan Carlos Ramirez, Sergio Sosa-Estani.

**Data curation:** Constanza Lopez-Albizu, Carolina Inés Cura.

**Formal analysis:** Andrés Benchetrit.

**Funding acquisition:** Sergio Sosa-Estani.

**Investigation:** Constanza Lopez-Albizu, Carolina Inés Cura, Juan Carlos Ramirez, Pamela Peyran, Emmaría Danesi.

**Methodology:** Constanza Lopez-Albizu, Carolina Inés Cura, Juan Carlos Ramirez.

**Project administration:** Constanza Lopez-Albizu, Carolina Inés Cura.

**Resources:** Sergio Sosa-Estani.

**Supervision:** Sergio Sosa-Estani.

**Validation:** Constanza Lopez-Albizu, Carolina Inés Cura, Juan Carlos Ramirez.

**Visualization:** Constanza Lopez-Albizu.

**Writing – original draft:** Constanza Lopez-Albizu.

**Writing – review & editing:** Constanza Lopez-Albizu, Carolina Inés Cura, Juan Carlos Ramirez, Pamela Peyran, Andrés Benchetrit, Emmaría Danesi, Sergio Sosa-Estani.

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
