## [Decision Letter · Decision Letter 0]

5 Oct 2024

Dear Dr. Lopez-Albizu,

Thank you very much for submitting your manuscript "Congenital Chagas Disease: A Cohort Study to Assess Molecular Diagnostic Methods at the Chagas Disease National Reference Center of Argentina" for consideration at PLOS Neglected Tropical Diseases. As with all papers reviewed by the journal, your manuscript was reviewed by members of the editorial board and two independent reviewers. In light of the reviews (below this email), we would like to invite the resubmission of a significantly-revised version that takes into account the reviewers' comments. 

Both reviewers provided favorable evaluations of the work. While reviewer #1 does not have many comments, reviewer #2 has raised several relevant comments that could help to improve the presentation of the manuscript. Both reviewers also indicated that figures could be improved. If you decide to submit a revised document, some attention should be given to methodological details and clarification of statistical analysis as indicated by reviewer #2. On your ethical statement, please include the number of the document approved by the Bioethics Committee. Note that the delay with this review process resulted from difficulties in identifying suitable reviewers for your work. I appreciate your patience, and should you have any questions, please do not hesitate to contact me.

We cannot make any decision about publication until we have seen the revised manuscript and your response to the reviewers' comments. Your revised manuscript is also likely to be sent to reviewers for further evaluation.

Sincerely,

Igor Cestari

Guest Editor

Laura-Isobel McCall

Section Editor

Reviewer's Responses to Questions

Key Review Criteria Required for Acceptance?

Methods

-Are the objectives of the study clearly articulated with a clear testable hypothesis stated?

-Is the study design appropriate to address the stated objectives?

-Is the population clearly described and appropriate for the hypothesis being tested?

-Is the sample size sufficient to ensure adequate power to address the hypothesis being tested?

-Were correct statistical analysis used to support conclusions?

-Are there concerns about ethical or regulatory requirements being met?

Reviewer #1: (No Response)

Reviewer #2: -Are the objectives of the study clearly articulated with a clear testable hypothesis stated?

Please define in a better way the objetives

-Is the study design appropriate to address the stated objectives?

-Is the population clearly described and appropriate for the hypothesis being tested?

Yes, It is. 

-Is the sample size sufficient to ensure adequate power to address the hypothesis being tested?

Yes, it is.

-Were correct statistical analysis used to support conclusions?

No, they were not. An ANOVA test results are mentioned in results section. Include and statistic analysis sections in the methodology. Did you evaluate the distribution of your data?. Could you include a Kappa Test to evaluate the results coherence between tests?

-Are there concerns about ethical or regulatory requirements being met?

More information regarding the 24% of patients who did not provide consent should be included. Additionally, could you provide a copy of the consent form as supplementary information?

Specific comments:

Introduction

Please include an explanation of the immune system in babies and why they may test negative after 11 months.

Methods

Consent for Patients: Only 24% of participants provided consent. Were they informed that their samples might be used for testing?

174: For IFI, until what antibody titer was considered positive, and what was the ELISA cutoff absorbance for positive an d negative ? Which positive and negative controls did you use? Did you use a TcVI strain of T. cruzi? Could this choice affect the detection of other DTUs, and what was the rationale behind selecting this strain? How many technical replicates were included?, Have you test your serological test against patients infected with other DTUs? 

175: Please provide a table with the test results for the mothers' T. cruzi testing, including antibody titers and ELISA outcomes.

Were DTUs detected in both mothers and children? If so, this could significantly enhance our understanding of T. cruzi transmission and epidemiology.

146: Were the samples collected in anticoagulant tubes? If so, please specify the type of anticoagulant used, or were they directly transferred to conservation solutions?

183: Please include the sequences of the primers and dyes used.

195: What was the CT cutoff value used to consider a sample positive? Please include the number of technical replicates used per sample.

241: Please specify the number of serial dilutions included for the curve. Also, describe the extraction procedure followed and provide references. Additionally, could you clarify why a TcVI strain was used for serological testing, while a TcV strain was used for the standard curve? Which parameter do you use to valid your curve. 

242: What controls were used for the Multiplex PCR? Did you include DNA for each DTU? How was the technique standardized in your lab? Please include this information in the paper, along with a detailed description of the PCR methodology.

Regarding the sensitivity and specificity of the test: How were these parameters evaluated?

Results

-Does the analysis presented match the analysis plan?

-Are the results clearly and completely presented?

-Are the figures (Tables, Images) of sufficient quality for clarity?

Reviewer #1: Are the figures (Tables, Images) of sufficient quality for clarity? The quality of the figures could be improved, especially those relating to supplementary data 1 and 2.

 All images that are colored red have poor resolution.

Reviewer #2: Does the analysis presented match the analysis plan?

The results section requires more comprehensive detail and in-depth analysis to enhance clarity and significance. Currently, the description of the results is minimal, which undermines the overall value of the paper. It is essential to present the findings in a more structured and informative manner, ensuring that critical data is highlighted and interpreted effectively.

-Are the results clearly and completely presented?

The current description of the results is insufficient for publication; a more detailed account is necessary. 

-Are the figures (Tables, Images) of sufficient quality for clarity?

No, as previously mentioned, additional analysis should be included. 

Specific comments

I believe you have valuable information to contribute to the field of congenital Chagas disease. However, the methodology you suggested is not clearly represented in the results section. To enhance the paper's acceptance, a thorough analysis, improved quality, and more informative figures should be included.

Details about the 499 children included in the study should encompass demographic information such as age, gender, and any relevant clinical history or maternal characteristics. Additionally, it would be useful to know whether they were breastfed or born via cesarian section. While the primary objective of the paper is to validate the qPCR techniques, this information could provide valuable context.

A detailed breakdown of the performance of the two qPCR techniques (qPCR SatDNA and qPCR kDNA) should include per each PCR:

• Specificity and sensitivity rates for both techniques. Please include details of ct, parasite load and limit of detection. 

• Any statistical analyses conducted to compare the qPCR results with the gold standard diagnostic algorithm (CCD-GS). Kappa tets could be useful. 

• Differences between PCR parasite loads.

Additionally, results concerning the performance of the CCD-GS algorithm should be provided as a comparison point for the qPCR methods. This may include the number of false positives and negatives encountered.

Statistical analyses that demonstrate the significance of the performance of the qPCR techniques, such as confidence intervals or p-values, should also be included.

Regarding the relationship between maternal DTU and babies, further clarification is needed.

I am wondering about table 2. What do you have decrease in the parasite load during the time 2?

Conclusions

-Are the conclusions supported by the data presented?

-Are the limitations of analysis clearly described?

-Do the authors discuss how these data can be helpful to advance our understanding of the topic under study?

-Is public health relevance addressed?

Reviewer #1: (No Response)

Reviewer #2: -Are the conclusions supported by the data presented?

-Are the limitations of analysis clearly described?

No, please include them.

-Do the authors discuss how these data can be helpful to advance our understanding of the topic under study?

No, a better conclusion should be include.

-Is public health relevance addressed?

The work addresses a significant public health issue; however, the results are not presented in a manner that reflects this importance.

Editorial and Data Presentation Modifications?

Reviewer #1: (No Response)

Reviewer #2: A major revision is required for the results, discussion, and conclusion sections. As I mentioned before, I consider the information from this work and the patient population to be valuable; however, the results are unspecified and poorly presented. Clearer and more detailed reporting of the findings is essential to enhance the overall quality of the paper.

Summary and General Comments

Reviewer #1: (No Response)

Reviewer #2: The study addresses a significant public health issue concerning congenital Chagas disease, emphasizing the importance of accurate diagnostic techniques. However, substantial revisions are needed in several areas to improve clarity and depth. The study presents valuable data regarding of qPCR techniques for diagnosing congenital Chagas disease, which could enhance testing and treatment strategies. Nonetheless, the results section is insufficiently detailed and lacks comprehensive analysis, undermining the overall significance of the findings. More thorough presentation of data is necessary, including demographic information of the study population and detailed performance metrics for qPCR methods. Additionally, the statistical analyses employed are not clearly articulated, and additional methods should be included to evaluate the coherence of results between different tests. The study has the potential to contribute significantly to the understanding of congenital Chagas disease transmission and epidemiology, but this potential is not fully realized in the current presentation. A more structured and informative approach is needed in the results and discussion sections. Furthermore, limitations of the study should be clearly stated, and the public health relevance of the findings must be emphasized more effectively. Overall, while the paper holds valuable information, it requires major revisions to meet publication standards.
---

## [Editor Report · Decision Letter 1]

16 Dec 2024

Dear Dr Lopez-Albizu,

We are pleased to inform you that your manuscript 'Congenital Chagas Disease: A Cohort Study to Assess Molecular Diagnostic Methods at the Chagas Disease National Reference Center of Argentina' has been provisionally accepted for publication in PLOS Neglected Tropical Diseases.

Best regards,

Igor Cestari

Guest Editor

Laura-Isobel McCall

Section Editor

Shaden Kamhawi

co-Editor-in-Chief

Paul Brindley

co-Editor-in-Chief

<style type="text/css">p.p1 {margin: 0.0px 0.0px 0.0px 0.0px; line-height: 16.0px; font: 14.0px Arial; color: #323333; -webkit-text-stroke: #323333}span.s1 {font-kerning: none

</style>

---

## [Editor Report · Acceptance letter]

2 Jan 2025

Dear Mg., Bq. Lopez-Albizu,

We are delighted to inform you that your manuscript, "Congenital Chagas Disease: A Cohort Study to Assess Molecular Diagnostic Methods at the Chagas Disease National Reference Center of Argentina," has been formally accepted for publication in PLOS Neglected Tropical Diseases.

Best regards,

Shaden Kamhawi

co-Editor-in-Chief

Paul Brindley

co-Editor-in-Chief
